# New perspectives on stress and negative emotions: Positive effects on adolescent learning, memory, and mental health

Jiawei Gao[1☉], Bohao Liu[2,3☉]*

1 Department of Philosophy, Sun Yat-sen University, China, 2 Department of Thoracic Surgery, The First Affiliated Hospital of Xi'an Jiaotong University, Xi'an China, 3 Key Laboratory of Enhanced Recovery After Surgery of Integrated Chinese and Western Medicine, Administration of Traditional Chinese Medicine of Shaanxi Province, The First Affiliated Hospital of Xi'an Jiaotong University, Xi'an, China

☉ These authors contributed equally to this work.
* lbh1001@stu.xjtu.edu.cn

## Abstract

### Background

Negative emotions have long been regarded as detrimental to learning and memory, while the potential benefits of moderate stress remain underexplored. This study investigates how moderate stress influences adolescent learning and memory, advocating for an assessment of emotions to provide novel theoretical insights into adolescent mental health.

### Methods

We recruited 53 middle school students and established a murine stress model. Behavioral tests and physiological indicators were systematically analyzed to evaluate the effects of varying stress intensities on learning, memory, and psychological states. Methods included acute stress tests, short-term memory assessments, and measurements of HPA axis hormone levels.

### Results

The moderate stress group exhibited significantly better memory performance than the control group, whereas the high-stress group showed marked memory decline. Murine experiments revealed that moderate stress enhanced learning efficiency and was associated with normal HPA axis hormone regulation, supporting the complexity of stress effects.

### Conclusions

Moderate stress can enhance adolescent learning and memory, challenging traditional views of stress as purely negative. Theoretically, moderate stress may activate

**Data availability statement:** All relevant data are within the manuscript and its Supporting information files.

**Funding:** The author(s) received no specific funding for this work.

**Competing interests:** The authors have declared that no competing interests exist.

adaptive mechanisms and improve cognitive function, offering a new framework for mental health interventions. These findings underscore the importance of balanced stress management in educational practices and psychological strategies.

## 1. Introduction

Emotions play a pivotal role in shaping adolescent mental health. Traditional cognitive theories often categorize emotions into a binary framework of "positive" versus "negative," assigning value based on this oversimplified dichotomy. Positive emotions—such as gratitude, pride, and joy—are typically associated with progress toward goals or external validation. As Meng [1] posits, positive emotions coordinate cognitive behaviors to fulfill diverse needs. Conversely, negative emotions, defined as affective states with negative valence, arise from internal or external factors that hinder task completion or rational thinking. Common examples include sorrow, anger, stress, fear, and hatred. For example, in their seminal and widely cited work on stress, Lazarus and Folkman [2] conducted a historical survey of research on the topic. Their analysis demonstrates that, despite terminological variations across disciplines—such as the preference for strain in sociology and anxiety in psychology—the core phenomenon under investigation remains essentially the same. In physiology, sociology, and psychology alike, stress is generally conceptualized as an internal state of disequilibrium or disruption resulting from interactions with changing external environmental conditions. During adolescence, negative emotions—manifesting as subjective experiences of distress—can disrupt learning processes and impair cognitive performance. Consequently, scholars have sought strategies to mitigate these emotions to enhance mental well-being. For instance, Sun et al [3] demonstrated that group counseling alleviates anxiety, while He [4] established peer support groups to reduce academic stress in middle school students. Shackman et al. [5] highlighted anxiety's detrimental impact on spatial memory.

However, as natural responses to external stimuli, negative emotions are integral to the emotional spectrum. Overemphasis on suppressing negative emotions in favor of positivity risks undermining adaptive coping mechanisms, ultimately harming mental health. Russell [6] challenged this dichotomy with the circumplex model, which evaluates emotions along two dimensions: valence (pleasure-displeasure) and arousal (activation-deactivation). While negative emotions are unpleasant, they can also motivate goal-directed behavior by prompting individuals to exert greater effort, thereby fostering achievement. Scholars like Greenberg [7] argue for the "functional utility" of negative emotions, and Tamir et al. [8] emphasize their "instrumental benefits," suggesting that individuals may deliberately endure short-term discomfort to enhance task engagement and achieve long-term goals.

Adolescents experience heightened emotional volatility due to neuroendocrine reorganization during puberty. As detailed in Smith et al. [9] interactions between stress-responsive hormones (CRH, GnRH, dopamine, serotonin) and psychosocial stressors amplify sensitivity to environmental challenges, resulting in increased

emotional lability. As they transition from childhood to adulthood, academic pressures often dominate, creating a vicious cycle: stress-induced negative emotions reduce learning efficiency, exacerbating both academic and psychological burdens. Thus, understanding and managing stress-related negative emotions is critical. Recent studies explore whether such emotions might paradoxically enhance mental resilience and learning efficiency. For example, Beste [10] found that stress improves multitasking efficiency, while Vogel [11] showed that acute stress enhanced learning from explicit instruction in a study with 61 participants. They were exposed to either a standardized stressor (the Socially Evaluated Cold Pressor Test) or a control condition before learning stimulus-response associations. Half of the associations were taught through direct instruction, while the others were learned by trial-and-error. Stress selectively improved performance on the instructed items, and this effect was linked to increased cortisol levels and autonomic nervous system activity. These findings underscore the need to investigate how stress-driven negative emotions influence adolescent memory and mental health.

This research recruited 53 middle school students to evaluate the effects of acute stress on short-term memory and psychological states. A murine stress model was also established to assess memory performance, anxiety/depression-like behaviors, and HPA-axis hormone levels under varying stress intensities. Cross-species approaches bridge complementary limitations: rodents enable precise neural manipulation but lack subjective emotional reporting, while human studies face physiological monitoring constraints during high-stress scenarios. Critically, research demonstrates that despite behavioral differences, episodic memory across mammals depends on conserved hippocampal-prefrontal circuitry [12,13]. Results revealed that moderate stress-induced anxiety enhanced learning and memory without compromising mental health. However, excessive stress disrupted corticosterone and ACTH secretion, potentially leading to pathological anxiety.

## 2. Methods

### 2.1 Recruitment and testing of adolescent participants

This study was conducted in compliance with ethical standards and received approval from the Ethics Committee of the First Affiliated Hospital of Xi'an Jiaotong University (XJTU1AF2022LSYY-40).

From May 2022 to August 2022, a total of 53 high school students (aged 18–19) who were in their final year of high school were recruited for participation in the research. Informed consent forms were distributed to all participants, and written consent was obtained from each participant prior to their involvement in the study. The informed consent process ensured that participants were fully aware of the study's nature, purpose, procedures, and potential risks. Participants were also given the opportunity to ask questions and receive further clarification. Participants were classified into categories A, B, C, or D based exclusively on their average score across three academic tests. Specifically, an average score between 562 and 750 (inclusive) defined category A, a score between 375 and 561 defined category B, a score between 187 and 374 defined category C, and a score between 0 and 186 defined category D.

The study involved random assignment of participants into three groups: a control group (n = 18), a mild-stress intervention group (n = 18), and a high-stress intervention group (n = 17). The experimental design comprised pre-test and post-test phases conducted sequentially within one hour.

Word Lists: Two lists of 15 Chinese words each (e.g., umbrella, poster, wallet) were derived from Madan's lexicon and adapted to common Chinese contexts. Pre-test Procedure: Memory Encoding Phase: Participants memorized the first word list in a closed, unproctored room. Each word was displayed centrally on a screen for 5 seconds, followed by a 3-second interval before the next word appeared.

Testing Phase: Participants sequentially input recalled words on a smartphone. Skipping was allowed for forgotten words. The test is automatically terminated after 120 seconds or upon final submission. Accuracy and order of responses determined scores.

Physiological and Psychological Measures: Heart rate variability (HRV) and anxiety scores (11-point scale: 0 = no anxiety, 10 = extreme anxiety) were recorded post-test.

Post-test Procedure: Control Group: Identical conditions to pre-test. Mild-Stress Group: Participants were informed of potential proctor observation. High-Stress Group: Tested under direct proctor supervision with an on-screen countdown. Post-test Measures: Participants completed the DASS-21 self-assessment, repeated anxiety scoring, and HRV recording (Fig 1).

## 2.2 Animal grouping

Thirty-six male C57BL/6J mice, aged 6–8 weeks, were purchased from the Xi'an Jiaotong University Animal Center. The mice were housed in a controlled environment with regulated temperature and lighting, and were provided ad libitum access to food and water. Behavioral testing, including the Open Field, Light/Dark Box Avoidance, and Elevated Plus Maze tasks, was conducted in compliance with the International Brain Research Organization (IBRO) ethics guidelines. For the Open Field test, the illumination was kept under 100 lux, with a maximum session duration of 10 minutes. In the Light/Dark Box, mild foot shocks (0.3 mA, 1-second duration) were applied with insulated floors to minimize discomfort, while in the Elevated Plus Maze, the arm width was 5 cm and the side walls were 15 cm to ensure safety and reduce stress. This study adhered to ethical principles regarding the humane treatment of animals, with every effort made to minimize pain, distress, and suffering. Terminal blood collection was performed under isoflurane anesthesia via cardiac

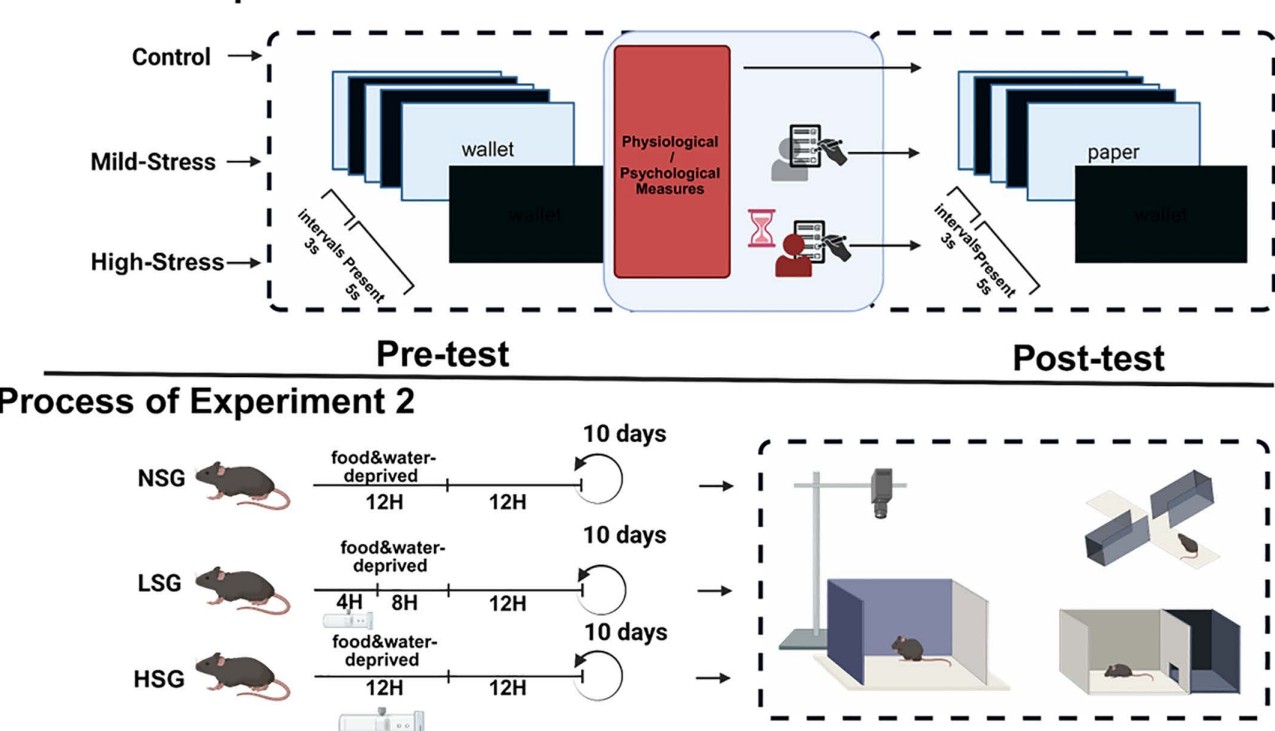

**Fig 1. Experimental workflow for stress and memory testing in humans and mice.** Process of experiment 1:Pre-test/post-test design with Control, Mild-Stress, and High-Stress conditions. Participants completed memory challenges with varying intervals before key assessments. Process of experiment 2:10-day protocols with differential deprivation periods: NSG, LSG (mixed 4H/8H deprivation), HSG (12H deprivation). All groups underwent behavioral testing.

puncture, followed by cervical dislocation, in accordance with AVMA guidelines. Serum samples were immediately processed to prevent hemolysis. The study protocol was approved by the Biomedical Ethics Committee of Xi'an Jiaotong University (XJTUAE2023−1697).

## 2.3 Restraint stress model

Restraint stress was performed 4/12h without access of food or water every day, according to the previous description with slight modification [14–16].Mice were randomly divided into three groups (n = 12 each):High-Stress Group (HSG): 12-hour daily restraint for 10 days.Low-Stress Group (LSG): 4-hour daily restraint for 10 days.Control Group (NSG): No restraint; housed individually.All groups were food- and water-deprived during restraint periods. NSG underwent deprivation without restraint. Mice were euthanized via orbital blood collection one day post-experiment (Fig 1).

## 2.4 ELISA

Plasma was isolated from whole blood via centrifugation. Corticosterone levels were measured using a Corticosterone ELISA Kit (Beyotime PC100), and ACTH levels were quantified with a Mouse ACTH ELISA Kit (Jianglai Biotech JL12373-96T).

## 2.5 Behavioral tests

Open Field Test:Mice were placed in a sanitized (75% ethanol) open-field arena for 5 minutes. Parameters recorded: average speed, central area path percentage, and central zone duration.

Elevated Plus Maze:Mice explored the maze for 5 minutes. Metrics included open/closed arm entries and time spent in each arm.

## 2.6 Passive avoidance memory test

A two-chamber apparatus (light/dark) with a 3.0 cm connecting hole was used. During training:Mice acclimated to the light chamber for 3 minutes.Upon entering the dark chamber (electrified floor), mice received a foot shock (5-minute session). After 24 hours, latency to re-enter the dark chamber and number of entries within 5 minutes were recorded. Unentered trials were scored as 300 s latency and 0 entries into the dark chamber.

## 3. Results

### 3.1 Effects of stress interventions on adolescent memory

All 53 participants completed the study. DASS-21 scores indicated normal ranges (anxiety <4, depression <5, stress <8). Gender distribution (male: 60.38%, female: 39.62%) and baseline anxiety/HRV scores showed no significant differences across groups (p > 0.05, Table 1).

Heart rate variability (HRV), defined as variations in heartbeat intervals, reflects autonomic nervous system regulation of cardiac function. Anxiety self-assessment scores provided subjective measures of participants' real-time anxiety and stress levels. Comparisons of pre- and post-intervention anxiety and HRV changes revealed that both mild- and high-stress interventions significantly increased anxiety, with the high-stress group showing the most pronounced changes (p < 0.001). These findings were corroborated by HRV metrics (Fig 2A and 2B).Correlation analyses indicated a significant positive relationship between anxiety and HRV changes in both stress intervention groups (p < 0.05, Fig 2C and 2D).Baseline memory scores showed no significant differences across groups (p = 0.720, η² = 0.013 , Table 2), indicating comparable pre-intervention memory abilities. Post-intervention score changes, however, differed significantly (p < 0.001,η² = 0.548).

We next performed a 2 (Time: Pre/Post) × 3 (Group: Control/Mild-Stress/High-Stress) repeated-measures ANOVA on participants' memory sequence scores. As shown in Fig 3A, results revealed significant main effects of both Time [$F_{(1,50)}$

**Table 1. Baseline characteristics of participants.**

| Characteristic | Control (n = 18) | Mild-Stress (n = 18) | High-Stress (n = 17) | Statistic | p-value | Effect |
|---|---|---|---|---|---|---|
| Gender[n (%)] | | | | $\chi^2 = 0.856$ | 0.312 | $\varphi c = 0.108$ |
| Male | 10 (55.56) | 11 (61.11) | 11 (64.71) | | | |
| Female | 8 (44.44) | 7 (38.89) | 6 (35.29) | | | |
| Age(Mean±SE) | 18.22 ± 0.10 | 18.11 ± 0.08 | 18.18 ± 0.13 | $F = 0.300$ | 0.742 | $\eta^2 = 0.012$ |
| Grade [n (%)] | | | | $\chi^2 = 4.436$ | 0.350 | $\varphi c = 0.289$ |
| A | 1(55.56) | 2(11.11) | 2(11.76) | | | |
| B | 16(88.89) | 15(83.33) | 11(64.71) | | | |
| C | 1(55.56) | 1(55.56) | 4(23.53) | | | |
| Anxiety Score (Mean±SE) | 3.12 ± 0.43 | 2.78 ± 0.41 | 9.59 ± 0.28 | $F = 0.070$ | 0.929 | $\eta^2 = 0.003$ |
| HRV (ms) | 76.94 ± 3.13 | 80.22 ± 2.78 | 80.29 ± 3.52 | $F = 0.374$ | 0.69 | $\eta^2 = 0.015$ |

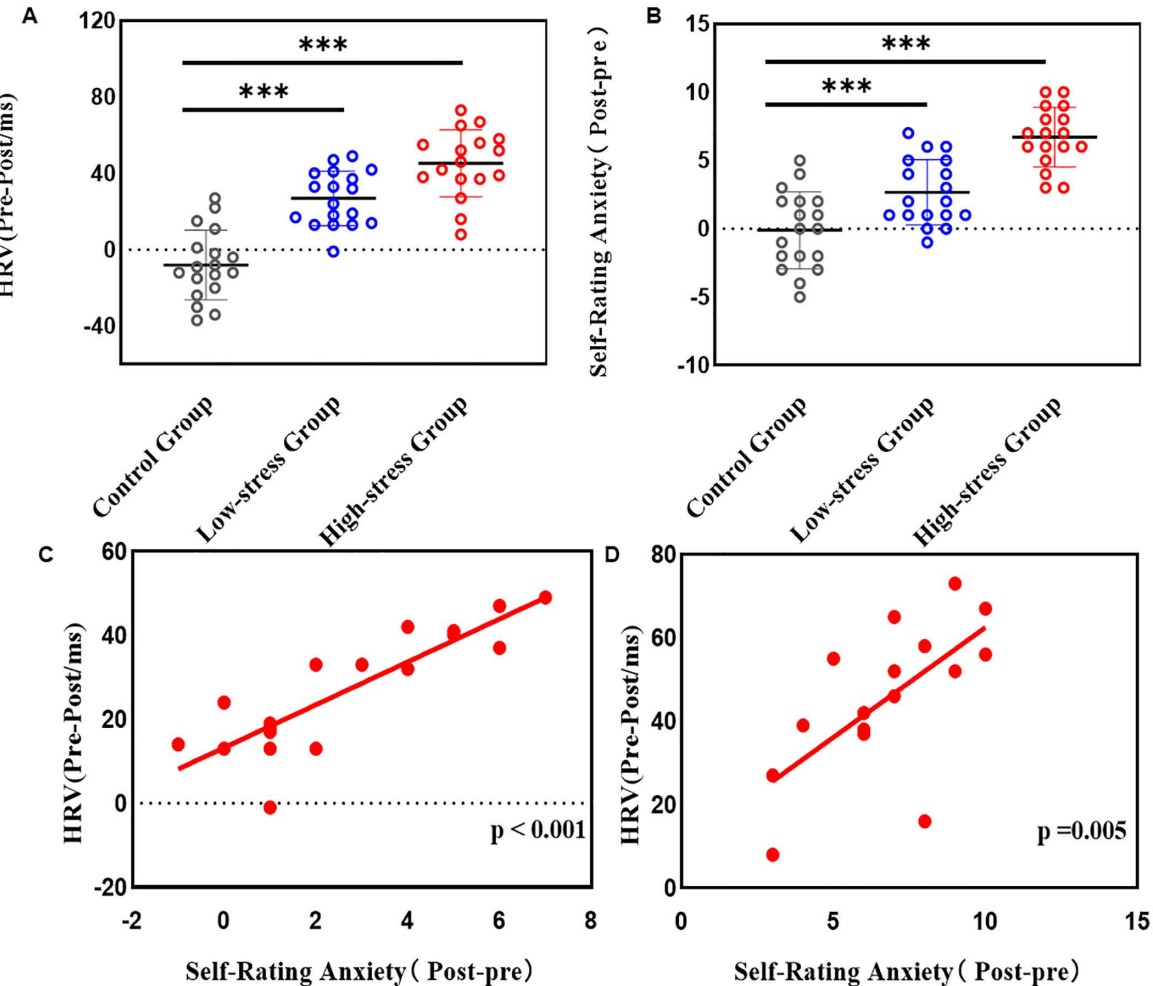

**Fig 2. Stress-level-dependent autonomic arousal and anxiety dynamics.** (A) Post-intervention HRV changes: Control (−8.0 ms), Mild-Stress (+26.9 ms), High-Stress (+45.2 ms). (B) Anxiety score shifts: Control (−0.11), Mild-Stress (+2.67), High-Stress (+46.71). (C) Mild-Stress group showed strong positive HRV-anxiety covariance (Pearson's $r = 0.859$, $p < 0.001$). (D) High-Stress group maintained significant but attenuated correlation ($r = 0.653$, $p = 0.005$). Dashed red lines = regression fits.

**Table 2. Memory scores across groups.**

| Parameter | Control (n=18) (Mean±SE) | Mild-Stress (n=18) (Mean±SE) | High-Stress (n=17) (Mean±SE) | F | p | η² |
|---|---|---|---|---|---|---|
| Pre-test | 6.50±0.31 | 6.67±0.35 | 6.29±0.31 | 0.33 | 0.72 | 0.013 |
| Post-Pre Δ | 3.28±0.55 | 6.11±0.54* | 0.35±0.46* | 30.29 | <0.001 | 0.548 |

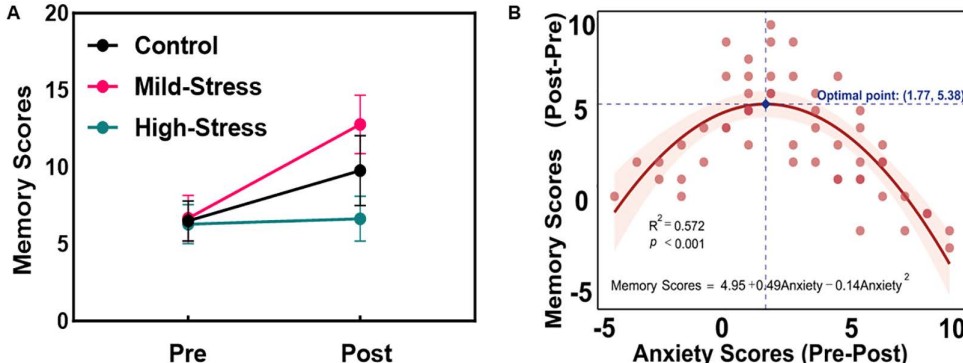

**Fig 3. Differential effects of stress interventions on sequence memory and the relationship between anxiety and memory.** (A) A repeated-measures ANOVA of memory sequence scores revealed a significant Time×Group interaction (F(2,72) = 30.29, p<0.001, η²=0.548, BF₁₀>100). The Mild-Stress condition enhanced post-training performance compared to the Control group (p<0.001), while the High-Stress condition impaired performance (p<0.001). Error bars represent ±1 standard error (SE).(B) Quadratic regression analysis demonstrated an inverted U-shaped relationship between changes in anxiety and improvements in memory (β_quad=−0.14, p<0.001; R²=0.572). The peak memory gain (+5.38 points, 95% CI: 4.60–6.16) was observed at an anxiety change of 1.77 (indicated by the dashed vertical line).

= 116.70, p<0.001, η²=0.700] and Group [F(2,50) = 29.92, p<0.001, η²=0.545, BF₁₀>100], along with a significant Time×Group interaction [F(2,72) = 30.29, p<0.001, η²=0.548, BF₁₀>100]. Post hoc analyses indicated no significant differences in pre-training test scores among the three groups, suggesting comparable baseline sequence memory capacity. However, after training, the Mild-Stress group showed significantly higher scores than the Control group (p<0.001), whereas the High-Stress group demonstrated significantly lower scores than Controls (p<0.001), indicating that stress interventions exert opposing effects on sequence memory.

To characterize the nonlinear relationship between anxiety and memory performance, we implemented a quadratic regression model (Fig 3B). This analysis revealed a significant inverted U-shaped relationship between anxiety change and memory change scores (quadratic term β=−0.14, p<0.001). The model explained 57.2% of the variance (R²=0.572, F(2,50) = 33.38, p<0.001). Peak memory performance occurred at an anxiety change score of 1.77, corresponding to a memory improvement of +5.38 points (95% CI: 4.60–6.16). Beyond this threshold, memory performance decreased significantly with increasing anxiety.

### 3.2 Effects of stress on murine emotional behavior

Open Field Test (OFT):

ANOVA on locomotion speed revealed significant group differences (F(2,33) = 40.76, p<0.001, η²=0.712). Post-hoc Dunnett's tests indicated mice in HSG displayed significantly increased locomotion speed compared to the NSG (10.62 vs. 8.11 cm/s; p<0.001), consistent with anxiety-like hyperactivity. Locomotion speed did not differ significantly between LSG and NSG (p=0.743) (Fig 4A).

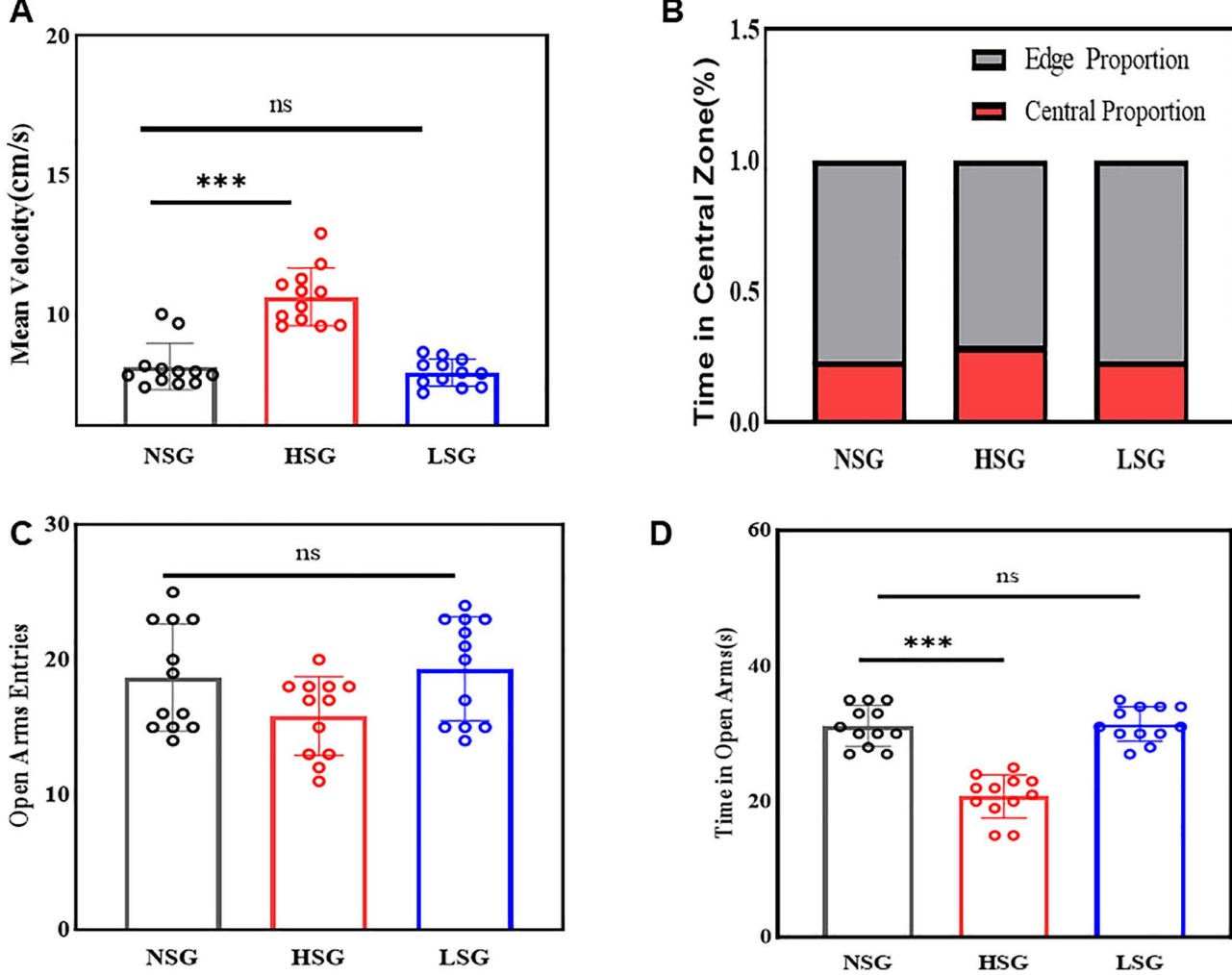

**Fig 4. Anxiety-like behavioral profiles in the OFT and EPM.** (A) Locomotion speed (OFT): A significant group difference was observed ($F_{(2,33)}$ = 40.76, $p < 0.001$, $\eta^2 = 0.712$). The HSG exhibited a higher locomotion speed compared to the NSG (10.62 cm/s vs. 8.11 cm/s; $p < 0.001$). No significant difference was found between the LSG and NSG ($p = 0.743$).(B) Central zone exploration percentage (OFT): A significant group difference was noted ($F_{(2,33)}$ = 27.704, $p < 0.001$, $\eta^2 = 0.627$). HSG demonstrated greater exploration in the central zone (28.97%) compared to both LSG (23.33%) and NSG (23.26%; both $p < 0.001$).(C) Open arm entries (EPM): No significant group difference was found ($F_{(2,33)}$ = 3.052, $p = 0.061$, $\eta^2 = 0.156$).(D) Time spent in open arms (EPM): A significant group difference was observed ($F_{(2,33)}$ = 52.534, $p < 0.001$, $\eta^2 = 0.761$). HSG spent less time in the open arms compared to NSG (20.75 s vs. 31.17 s; $p < 0.001$). No significant differences were noted between LSG and NSG for both entries ($p = 0.195$) and duration ($p = 0.969$).

ANOVA for central zone exploration distance percentage also showed significant intergroup variation ($F_{(2,33)}$ = 27.704, $p < 0.001$, $\eta^2 = 0.627$). The HSG exhibited higher central exploration (28.97%) than both LSG (23.33%) and NSG (23.26%; $p < 0.001$) (Fig 4B). However, this finding requires cautious interpretation, as the HSG group also demonstrated increased locomotor activity (10.62 cm/s vs. 8.11 cm/s). Consequently, the elevated central exploration observed in the HSG group may reflect heightened locomotor activity rather than a direct indication of reduced anxiety levels.

Elevated Plus Maze (EPM):

Analysis of open arm entries revealed no statistically significant differences across groups ($F_{(2,33)}$ = 3.052, $p = 0.061$, $\eta^2 = 0.156$). Though numerically lower in the HSG (15.83 entries) compared to NSG (18.67 entries; $p = 0.113$), this reduction did not reach statistical significance. For open arm duration, significant group differences emerged ($F_{(2,33)}$ = 52.534,

p<0.001, η²=0.761), with HSG spending significantly less time than NSG (20.75s vs. 31.17s; p<0.001). LSG showed no differences versus NSG in either entries (19.33 vs. 18.67; p=0.195) or duration (31.42s vs. 31.17s; p=0.969) (Fig 4C and 4D). The suppressed open arm exploration in HSG persisted despite generalized hyperactivity, supporting an anxiety-like phenotype.

### 3.3 Effects of stress on murine learning and memory

No statistically significant differences were found in dark chamber entries among groups (F(2,33) = 2.785, p=0.076, η²=0.144). Numerically, LSG exhibited fewer entries (0.50) compared to NSG (0.58) and HSG (1.00), though these differences did not reach statistical significance (p>0.05 for all pairwise comparisons) (Fig 5A).

For latency to re-enter the dark chamber, intergroup variation was significant (F(2,33) = 11.808, p<0.001, η²=417). LSG demonstrated prolonged latency compared to NSG (280.8s vs. 211.7s; p=0.029), while HSG showed shortened latency (135.0 s vs. 211.7 s; p=0.015) (Fig 5B).

Significant differences in dark chamber occupancy among the groups were identified using one-way ANOVA (F(2,33) = 12.416, p<0.001, η²=0.429) (Fig 5C). Post-hoc Dunnett's tests indicated that HSG spent significantly more time in the dark chamber compared to the NSG (18.6 s vs. 3.3 s; p<0.001). In contrast, no significant difference was found between the LSG and NSG (1.58 s vs. 3.3 s; p=0.685). Moreover, HSG exhibited a significantly longer duration in the dark chamber than LSG (p<0.001), suggesting that alterations in spatial avoidance behavior are dependent on stress intensity.

### 3.4 Effects of stress on HPA axis hormones

Stress exposure triggered hierarchical HPA axis activation, beginning with hypothalamic corticotropin-releasing hormone (CRH) release, followed by pituitary ACTH secretion, ultimately elevating adrenal corticosterone production. Serum corticosterone levels differed significantly across groups (F(2,33) = 41.727, p<0.001, η²=0.717). Post-hoc tests showed HSG exhibited markedly elevated corticosterone versus NSG (91.32 vs. 58.14 ng/mL; p<0.001), while LSG displayed intermediate elevation (68.82 vs. 58.14 ng/mL; p=0.012). For ACTH, significant intergroup variation emerged (F(2,33) = 45.066, p<0.001, η²=0.732), with HSG showing drastic increases over NSG (180.2 vs. 159.1 pg/mL; p<0.001), whereas LSG remained unchanged (155.3 vs.159.1 pg/mL; p=0.305) (Fig 6).

## 4. Discussion

This study investigated the potential positive association between moderate psychological stress and adolescents' learning performance, particularly with regard to memory outcomes. Our empirical findings suggest that moderate levels of stress do not significantly impair adolescents' psychological well-being, and may, in fact, enhance certain aspects of cognitive functioning, such as memory performance. However, it is important to recognize the limitations of purely behavioral and quantitative measures in capturing the experiential depth of stress and memory. Stress, as an internal and often pre-reflective experience, exerts regulatory influences on subjective activity in ways that may not be outwardly visible. Likewise, memory, as a conscious process, fundamentally involves the dynamic allocation of attention—an aspect that cannot be fully accounted for by observable behaviors or numerical indices.

From this perspective, phenomenological analysis offers a valuable framework for uncovering the underlying structures of consciousness that inform and shape such experiences. By remaining faithful to the first-person dimension of experience, phenomenology not only provides a deeper interpretation of the empirical results but also offers conceptual guidance for the design of stress-inducing tasks in educational contexts. It thus bridges the gap between empirical findings and the lived experience of learners, contributing to both theoretical understanding and practical application.

Interestingly, Lazarus and Folkman had already recognized the affinity between phenomenology and psychology in the study of inner experience, and they explicitly incorporated phenomenological ideas into their frameworks of cognitive

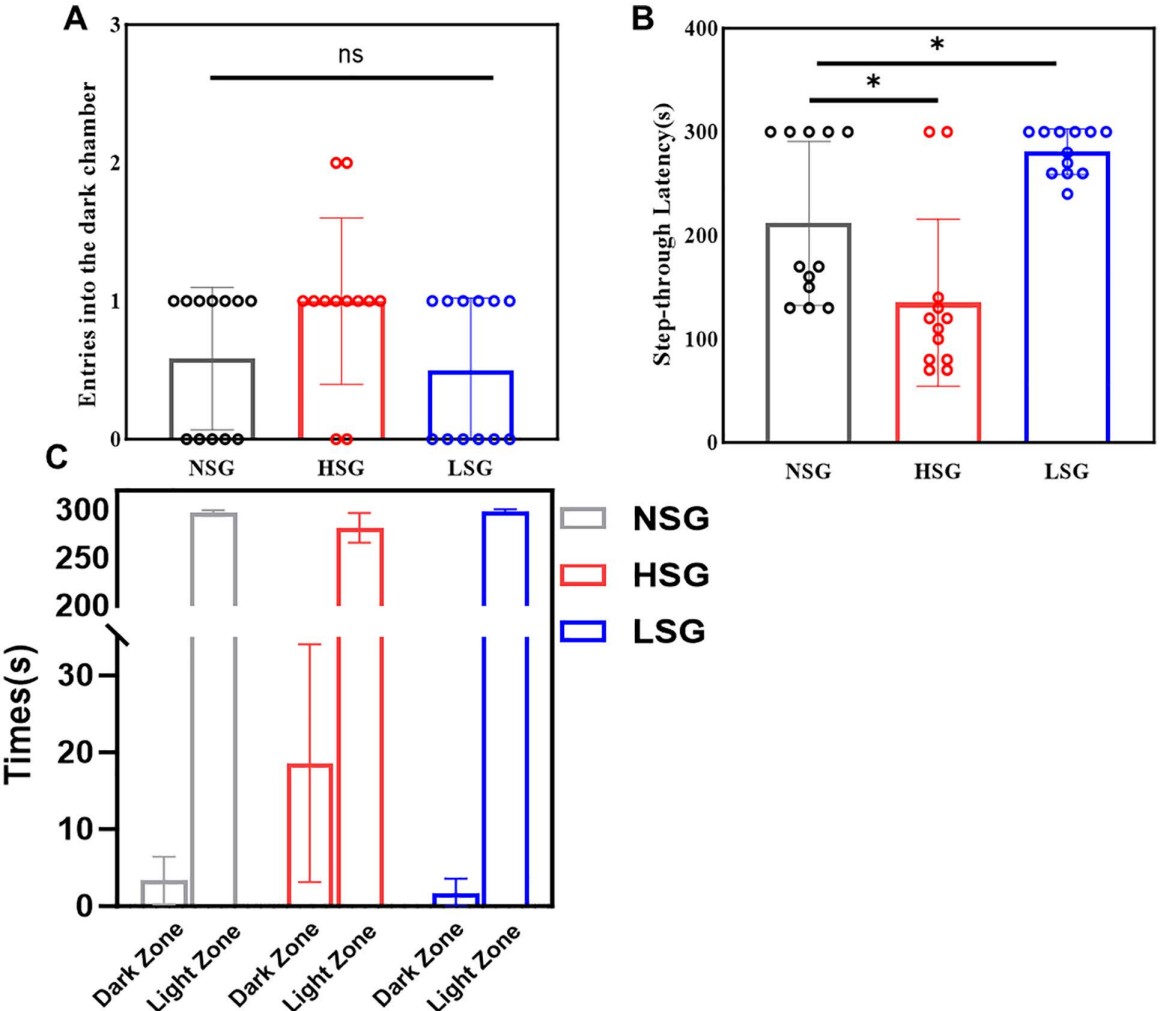

**Fig 5. Results of the Passive Avoidance Test across stress intensity groups.** (A) Dark chamber entries: No significant differences were observed among the groups (F(2,33) = 2.785, p = 0.076, η² = 0.144). The number of entries was as follows: NSG = 0.58, LSG = 0.50, HSG = 1.00 (all pairwise comparisons non-significant, p > 0.05). (B) Latency to re-enter the dark chamber: Significant group differences were found (F(2,33) = 11.808, p < 0.001, η² = 0.417). LSG showed a longer latency compared to NSG (280.8 s vs. 211.7 s; p = 0.029), while HSG exhibited a shorter latency than NSG (135.0 s vs. 211.7 s; p = 0.015).(C) Dark chamber occupancy time: Significant group differences were noted (F(2,33) = 12.416, p < 0.001, η² = 0.429). HSG spent more time in the dark chamber compared to NSG (18.6 s vs. 3.3 s; p < 0.001) and LSG (18.6 s vs. 1.58 s; p < 0.001). No significant difference was found between LSG and NSG (1.58 s vs. 3.3 s; p = 0.685).

appraisal and coping. This recognition affirms the relevance of phenomenology for stress research. However, their interpretation tends to reduce phenomenology to a form of "the individual's subjective interpretation of a transaction" [2], without fully engaging with its conceptual foundations and central commitments. For this reason, a more rigorous clarification of the phenomenological standpoint is necessary—one that provides a theoretically robust and experientially grounded account of how moderate stress interacts with attentional and mnemonic processes without resulting in psychological harm.

From a phenomenological perspective, conscious activity always unfolds within a certain horizon or background structure. This means that, in general, the intentional act that is actively carried out at the present moment carries the highest degree of attention and constitutes the focal point of consciousness, while other contents are perceived only marginally

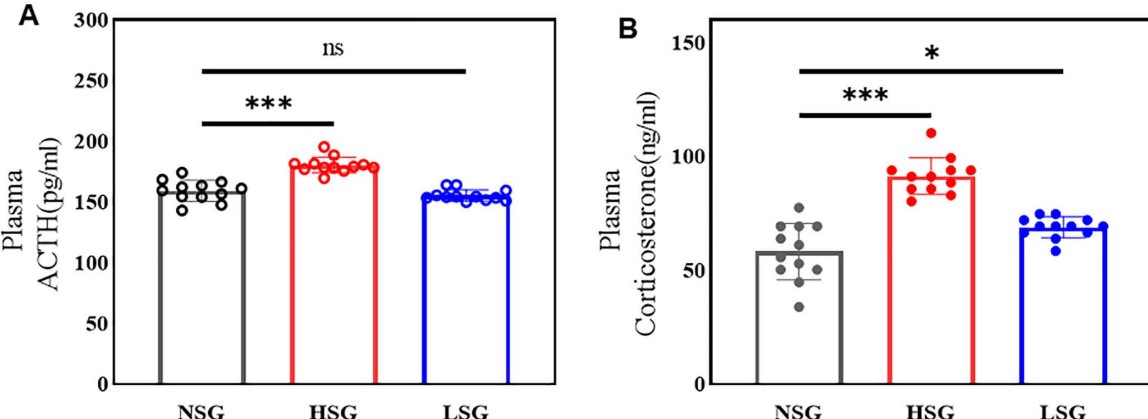

**Fig 6. Hormonal responses of the HPA axis to stress interventions.** (A) Plasma ACTH levels: Significant differences were observed among the groups ($F_{(2,33)}$ = 45.066, $p < 0.001$, $\eta^2 = 0.732$). The HSG exhibited higher ACTH levels compared to the NSG (180.2 pg/mL vs. 159.1 pg/mL; $p < 0.001$). No significant difference was found between the LSG and NSG (155.3 pg/mL vs. 159.1 pg/mL; $p = 0.305$).(B) Serum corticosterone levels: Significant group differences were noted ($F_{(2,33)}$ = 41.727, $p < 0.001$, $\eta^2 = 0.717$). HSG had elevated corticosterone levels compared to NSG (91.32 ng/mL vs. 58.14 ng/mL; $p < 0.001$), and LSG also showed higher levels than NSG (68.82 ng/mL vs. 58.14 ng/mL; $p = 0.012$).

or peripherally [17]. However, the surrounding sensory field continuously stimulates the subject, and once the contrast intensity of the environment reaches a certain threshold, it may trigger a shift in attention and lead the subject to reorient their focus [18].

In this study, environmental changes introduced during the post-test phase (e.g., the presence or absence of an invigilator, the appearance of a countdown timer on the screen) served as variables influencing participants' attention. Specifically, the three groups of middle school students were exposed to different environmental cues during the post-test, which induced different patterns of attentional behavior and led them to focus on different aspects of the situation. Compared to the control group, the low-stress intervention group was merely informed that they might be observed by an invigilator—an implicit and unrealized stressor. Although participants in this group became aware of a possible change in the test setting, they were not significantly distracted, since the change had not yet materialized. Their attention remained largely focused on the word memorization task.

In contrast, the high-stress group experienced a clear and present environmental shift, in which the contrast reached a threshold sufficient to redirect their conscious focus. As a result, participants in this group were no longer able to concentrate fully on the memory task; instead, their attention was drawn to the testing environment itself. The accompanying sense of pressure gradually intensified and ultimately may have overridden their cognitive engagement with the memory task, thereby impairing memory performance.

This interpretation is supported by the HRV data. As a physiological indicator of stress levels, HRV revealed no significant change in the control group, with a mean difference of –8 ms between pre- and post-test measurements. In the low-stress group, the mean difference increased to 26.89 ms, and in the high-stress group, it was the highest, reaching 43.94 ms. These results confirm the relationship between experienced stress and shift in attentional orientation.

Further correlation analyses revealed that stress fluctuations directly correlated with anxiety levels, which ultimately affected memory performance. Moderate anxiety exhibited functional benefits, as reflected in vocabulary gains: from 6.5 to 9.78 words in the control group, and from 6.67 to 12.78 in the low-stress group. However, the high-pressure group underperformed relative to controls, indicating that anxiety's instrumental benefits are threshold dependent. Previous studies align with this finding [19], showing that negative emotions can promote non-creative, rehearsal-based learning strategies.

Our study has several important limitations that warrant consideration. First, the exclusive use of male mice restricts our ability to examine sex-specific stress responses, highlighting the need for future research to include female cohorts controlled for estrous cycle phases, given the established sexual dimorphism in HPA-axis regulation—where estrogen modulates glucocorticoid receptor sensitivity [20,21]. Second, while we demonstrate parallel stress-memory effects across species, the direct mechanistic links between human verbal recall and rodent contextual memory remain unexplored. This gap underscores the necessity for advanced cross-species approaches, such as comparative fMRI and c-FOS mapping of hippocampal-prefrontal circuits, to elucidate shared neurobiological substrates. Additionally, while murine corticosterone levels provide valuable mechanistic insights, they should not be directly equated with human cortisol dynamics due to species-specific physiological differences. Our educational implications are further constrained by the single-center adolescent cohort, indicating a need for multi-center replications with larger, more diverse samples to enhance generalizability.

Despite a sample size of 53 participantsand a power analysis indicating over 99.9% statistical power, suggesting robustness in our findings, we did not collect data on socioeconomic status. Future research should aim to include broader demographics to better understand the impact of socioeconomic factors. Finally, while we utilized the number of entries into the dark chamber as a measure of memory recall and anxiety-like behavior, this approach has limitations, as it may also reflect altered risk assessment or differences in nociceptive sensitivity.

Through integrated human-adolescent and murine studies, we demonstrate that moderate stress enhances memory performance while severe stress impairs it across species. This manifests behaviorally as an inverted U-curve relationship between anxiety levels and memory outcomes—human vocabulary recall peaked at moderate anxiety. These findings align with established evidence that excessive stress disrupts hippocampus-dependent declarative memory in both humans (verbal recall) and rodents (spatial-contextual tasks) [22]. Critically, despite differing task modalities, conserved hippocampal-prefrontal circuitry underpins memory processing in both species [12,13], and human vocabulary strategies often engage contextual binding mechanisms analogous to rodent episodic encoding [23,24].But the physical restraint approach in mice and acute stress procedures in humans represent distinct experimental systems. Rodent restraint primarily focuses on physiological responses to sensory-motor confinement, while human stress testing emphasizes cognitive evaluation during social challenges. This distinction highlights that neither model fully replicates the other's stress characteristics—restraint effectively captures bodily stress reactions, whereas human testing reflects psychosocial dimensions. However, this divergence introduces inherent limitations: murine restraint fails to model anticipatory anxiety and social-evaluative components that are central to human psychological stress, potentially limiting its translational relevance for disorders such as test anxiety and performance phobias. To address these limitations, research strategies must recognize the unique contributions of each system. Mouse models allow for detailed monitoring of physiological changes under controlled conditions; yet their inability to replicate higher-order cognitive aspects of human stress remains a fundamental constraint. Conversely, human studies capture subjective experiences and complex behavioral adaptations that rodents cannot express, but they face ethical and methodological challenges in monitoring real-time neuroendocrine dynamics.

Our murine model further revealed HPA-axis dynamics wherein moderate stress preserved negative feedback glucocorticoids, whereas severe stress induced dysregulation. Although murine hormone levels cannot be directly extrapolated to humans, the conserved genetic architecture regulating anxiety suggests shared neuroendocrine principles [25]. Ultimately, cross-species approaches provide complementary insights: human studies capture subjective anxiety states and real-world stress responses, while murine models enable precise neural monitoring—yet neither fully explains the other. Future work should dissect stress-memory interactions through dual-species neural activity profiling (hippocampal theta coherence) to resolve mechanistic convergences beyond behavioral correlations.

## 5. Conclusion

Through human and murine experiments, this study reveals stress's complex effects on learning and memory, particularly highlighting moderate stress's positive impacts. These findings challenge traditional views of stress as purely detrimental,

emphasizing its potential to enhance cognitive function and psychological resilience. Theoretically, moderate anxiety may activate adaptive mechanisms that improve attention and information processing, likely mediated through HPA axis activation and hormonal changes.

Future research should explore individualized stress thresholds and their neurobiological underpinnings, advancing our understanding of stress-memory interactions. Through human and murine experiments, this study reveals stress's complex effects on learning and memory, particularly highlighting moderate stress's positive impacts. These findings challenge traditional views of stress as purely detrimental, emphasizing its potential to enhance cognitive function and psychological resilience. Theoretically, moderate anxiety may activate adaptive mechanisms that improve attention and information processing, likely mediated through HPA axis activation and hormonal changes.

## Supporting information

**S1 File. Original data for the all chart.**
(XLSX)

## Author contributions

**Conceptualization:** Bohao Liu.

**Data curation:** Bohao Liu.

**Methodology:** Bohao Liu.

**Supervision:** Bohao Liu.

**Writing – original draft:** Jiawei Gao, Bohao Liu.

**Writing – review & editing:** Jiawei Gao.

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
