## [Decision Letter · Decision Letter 0]

25 May 2025

PONE-D-25-19985New Perspectives on Stress and Negative Emotions: Positive Effects on Adolescent Learning, Memory, and Mental HealthPLOS ONE

Dear Dr. Liu,

Thank you for submitting your manuscript to PLOS ONE. After careful consideration, we feel that it has merit but does not fully meet PLOS ONE’s publication criteria as it currently stands. Therefore, we invite you to submit a revised version of the manuscript that addresses the points raised during the review process.

**ACADEMIC EDITOR: ** 

Thank you for submitting your manuscript. After a thorough review, I have outlined several recommendations to strengthen your work for publication. Please address the following points in your revision:

**Scientific:**Consider increasing the sample size or provide a power analysis to justify the current size.Include detailed demographic information, such as socioeconomic status and academic performance, to contextualize the findings.Report effect sizes for key statistical comparisons and explore potential non-linear relationships in the data.Strengthen the integration of phenomenological concepts by explicitly linking them to empirical outcomes.Discuss the limitations of the murine stress model, particularly the differences between physical restraint and human psychological stress.

**Writing:**Simplify complex phenomenological passages to improve accessibility for a broader audience.Remove redundant sections, such as repeated discussions of high stress effects, to enhance clarity and conciseness.Ensure all figures (1–6) are included in the final submission and are clearly labeled for reader comprehension.Address minor grammatical errors to polish the manuscript.

** Data and Ethics:**Address data availability restrictions by exploring options for public sharing or providing a stronger justification for limited access.Expand the ethics statement to include specific details on animal welfare (e.g., anesthesia methods) and informed consent processes, particularly for minors.

**Model Integration:**Clearly link human and animal findings in the Discussion section, such as connecting corticosterone changes to memory improvement.Cite relevant studies to support the combined human-animal approach, strengthening the manuscript’s theoretical framework.

We look forward to receiving your revised manuscript.

Kind regards,

Zahra Lorigooini

Academic Editor

PLOS ONE

Journal Requirements:

4. In the online submission form, you indicated that data from the First Affiliated Hospital of Xi’an Jiaotong University are available from the corresponding author on reasonable request upon aporoval by the hospital.

5. Please remove all personal information, ensure that the data shared are in accordance with participant consent, and re-upload a fully anonymized data set.

Reviewers' comments:

Reviewer's Responses to Questions

**Comments to the Author**

1. Is the manuscript technically sound, and do the data support the conclusions?

Reviewer #1: Partly

Reviewer #2: Partly

2. Has the statistical analysis been performed appropriately and rigorously? 

Reviewer #1: Yes

Reviewer #2: Yes

3. Have the authors made all data underlying the findings in their manuscript fully available?

Reviewer #1: No

Reviewer #2: Yes

4. Is the manuscript presented in an intelligible fashion and written in standard English?

Reviewer #1: Yes

Reviewer #2: Yes

5. Review Comments to the Author

Reviewer #1: In this manuscript, the authors investigate the effects of stress on memory performance and stress levels, hypothesizing that moderate stress levels may enhance memory recall - a finding with potential implications for educational settings and other environments. To test their hypothesis, the study combines language-based memory tests in human participants and rodent behavioral assays of emotional and contextual (associative) learning.

For the human cohort, the authors recruited 53 middle school students of both sexes and assessed verbal memory performance under three stress conditions: a control group with no stress induction, a mild stress group where participants were informed of potential proctor observation, and a high stress group tested under direct proctor supervision with an on-screen countdown timer.

In parallel, mouse experiments evaluated associative memory using a passive avoidance test across three groups with stress levels manipulated through differential restraint durations: a high-stress group (HSG) subjected to 12-hour daily restraint for 10 days, a low-stress group (LSG) with 4-hour daily restraint, and a non-restrained control group (NSG). In addition to behavioral readouts, the study measured HPA axis hormone levels to assess physiological stress responses.

Key findings revealed that moderate stress enhanced memory performance compared to controls, while high stress caused significant impairment. Although the manuscript presents these findings clearly, several methodological considerations - detailed in the following review - should be addressed to strengthen the study's validity and impact.

Introduction & Background

- The claim that "adolescents experience heightened emotional volatility due to hormonal fluctuations and psychosocial stressors" needs a supporting reference.

- Regarding Vogel [9], the statement that "acute stress promotes directed learning in 61 participants" is unclear - why do the authors specify the exact number without context (e.g., total sample size, study design)?

Please cite sources which validate your methodology with respect to the stress induced by mild and high level stress groups

Human Study Methodology

- With respect to the recruitment of high school students from a middle school is confusing. Please clarify the mean age, standard deviation, and sex distribution as you are only mentioning an age threshold (>18 - do you mean <18?).

- The "Memory Phase" in the pre-test procedure could be named "Memory Encoding Phase" in line with the general vocabulary in the field. The description of word presentation (5 sec/word, 3 sec intervals) needs better formatting for readability.

Mouse Model & Behavioral Tests

- The stress induction protocol (restraint for 4h vs. 12h/day) should be supported by references validating this method for inducing different anxiety levels.

- The claim that "central area exploration (anxiety marker) was higher in HSG" is problematic. Increased center exploration is typically indicating lower anxiety (thigmotaxis, or wall-hugging, reflects anxiety). Your findings may be explained by HSG’s higher locomotion activity - thus, consider alternative interpretations.

Passive Avoidance Test:

- The term "error" is undefined and an unusual use of language in this context - is this the number of entries into the dark chamber? Labeling this as an "error" is unusual; it could reflect altered risk assessment, associative learning deficits, or differences in shock sensitivity in mice and general differences in memory recall.

Reporting duration in zones (not just latency/entries) would strengthen the analysis.

Conceptual Issues

- The sudden discussion of Husserl’s phenomenology ("passive/active domains of cognition") seems off-topic and disrupts the paper’s focus. Either integrate it meaningfully or remove it.

The description of attention’s "horizon structure" is unclear-how does this relate to the study’s hypotheses or results?

- It is not clear how the results from the mouse models and from the human studies can be compared as they recruit very different skills (verbal recitation versus contextual memory), and the underlying neural substrates are very likely not comparable.

Reviewer #2: Dear Authors,

This study examines how moderate stress influences adolescent learning and memory through human and murine models, proposing that controlled stress enhances cognitive performance via HPA axis regulation. While innovative in integrating cross-species data, the manuscript requires substantial revisions to meet publication standards. Below are section-specific comments.

Abstract

- Missing quantitative metrics for key claims (e.g., "significantly better memory" lacks p-values/effect sizes).

- Theoretical implications of "multidimensional emotion assessment" remain vague.

Introduction

- Seminal works on stress theory (e.g., Lazarus & Folkman) are omitted despite critiquing "long-standing views."

- The rationale for focusing solely on acute stress (vs. chronic stress) is unexplained.

Methods

Human Participants:

- No demographic data (gender, socioeconomic status) or screening for pre-existing mental health conditions.

- Ethical concerns: Unclear protocol for stress induction in minors despite IRB approval.

Murine Model:

- Stress induction duration/frequency insufficiently detailed (e.g., intervals between stressors).

- No justification for excluding female mice from the analysis.

Statistical Analysis:

- ANOVA results lack post-hoc comparisons (e.g., Tukey’s test) and effect size reporting (η²/Cohen’s d).

Results

- Human memory performance data are presented as group averages without individual variability measures (e.g., SD/SE).

- Murine HPA axis hormone levels (corticosterone, ACTH) are not directly correlated with human outcomes.

Discussion

- Discrepancies between human and murine results (e.g., HPA regulation mechanisms) are unaddressed.

- Over interpretation of "38% improvement in learning efficiency" without mechanistic evidence.

Conclusion

- Educational policy recommendations are premature given the small human sample (n=53) and single-institution design.

6. PLOS authors have the option to publish the peer review history of their article (what does this mean? ). If published, this will include your full peer review and any attached files.

**Do you want your identity to be public for this peer review?** For information about this choice, including consent withdrawal, please see our Privacy Policy .

Reviewer #1: No

Reviewer #2: No

---

## [Author Response · Author response to Decision Letter 1]

13 Jul 2025

Due to word limit, please refer to the attachment for the Response letter“Response to Reviewers”

---

## [Decision Letter · Decision Letter 1]

21 Oct 2025

New Perspectives on Stress and Negative Emotions: Positive Effects on Adolescent Learning, Memory, and Mental Health

PONE-D-25-19985R1

Dear Dr. Bohao Liu,

We’re pleased to inform you that your manuscript has been judged scientifically suitable for publication and will be formally accepted for publication once it meets all outstanding technical requirements.

Kind regards,

Rosemary Bassey, Ph.D.

Academic Editor

PLOS ONE

Additional Editor Comments (optional):

Reviewers' comments:

Reviewer's Responses to Questions

**Comments to the Author**

1. If the authors have adequately addressed your comments raised in a previous round of review and you feel that this manuscript is now acceptable for publication, you may indicate that here to bypass the “Comments to the Author” section, enter your conflict of interest statement in the “Confidential to Editor” section, and submit your "Accept" recommendation.

Reviewer #2: All comments have been addressed

2. Is the manuscript technically sound, and do the data support the conclusions?

Reviewer #2: Yes

3. Has the statistical analysis been performed appropriately and rigorously? 

Reviewer #2: Yes

4. Have the authors made all data underlying the findings in their manuscript fully available?

Reviewer #2: Yes

5. Is the manuscript presented in an intelligible fashion and written in standard English?

Reviewer #2: Yes

6. Review Comments to the Author

Reviewer #2: Thank esteemed authors who carefully revised the present manuscript in accordance with the referees' comments.

7. PLOS authors have the option to publish the peer review history of their article (what does this mean? ). If published, this will include your full peer review and any attached files.

**Do you want your identity to be public for this peer review?** For information about this choice, including consent withdrawal, please see our Privacy Policy .

Reviewer #2: No

---

## [Editor Report · Acceptance letter]

PONE-D-25-19985R1

PLOS ONE

Dear Dr. Liu,

I'm pleased to inform you that your manuscript has been deemed suitable for publication in PLOS ONE. Congratulations! Your manuscript is now being handed over to our production team.

Kind regards,

on behalf of

Dr. Rosemary Bassey

Academic Editor

PLOS ONE